# ASFF-YOLOv5: Multielement Detection Method for Road Traffic in UAV Images Based on Multiscale Feature Fusion

Mulan Qiu [1], Liang Huang [1,2,*] and Bo-Hui Tang [1]

1   Faculty of Land Resources Engineering, Kunming University of Science and Technology,
    Kunming 650093, China; qiumulan@stu.kust.edu.cn (M.Q.); tangbh@kust.edu.cn (B.-H.T.)
2   Surveying and Mapping Geo-Informatics Technology Research Center on Plateau Mountains of Yunnan
    Higher Education, Kunming 650093, China
*   Correspondence: kmhuangliang@kust.edu.cn

**Abstract:** Road traffic elements are important components of roads and the main elements of structuring basic traffic geographic information databases. However, the following problems still exist in the detection and recognition of road traffic elements: dense elements, poor detection effect of multi-scale objects, and small objects being easily affected by occlusion factors. Therefore, an adaptive spatial feature fusion (ASFF) YOLOv5 network (ASFF-YOLOv5) was proposed for the automatic recognition and detection of multiple multiscale road traffic elements. First, the K-means++ algorithm was used to make clustering statistics on the range of multiscale road traffic elements, and the size of the candidate box suitable for the dataset was obtained. Then, a spatial pyramid pooling fast (SPPF) structure was used to improve the classification accuracy and speed while achieving richer feature information extraction. An ASFF strategy based on a receptive field block (RFB) was proposed to improve the feature scale invariance and enhance the detection effect of small objects. Finally, the experimental effect was evaluated by calculating the mean average precision (mAP). Experimental results showed that the mAP value of the proposed method was 93.1%, which is 19.2% higher than that of the original YOLOv5 model.

**Keywords:** object detection; road traffic multiple elements; adaptively spatial feature fusion; spatial pyramid pooling fast; basic traffic geographic information database

## 1. Introduction

Road traffic elements are important components of roads, which are the main contents for basic traffic geographic information database construction and are especially important for the development of basic traffic geographic information. Road traffic element information includes road centerlines, road intersections, zebra crossings, bus stations, roadside parking spaces, and other information [1]. Their accurate recognition and detection can provide important data support for automatic driving, improving intelligent transportation systems, promoting smart cities, and updating basic traffic geographic information databases. Currently, most research has focused on the detection and recognition of single-element traffic signs [2–4], extraction of road network information [5–8], real-time monitoring of road conditions [9,10], etc. while there are few studies on the extraction of road traffic multielement information, and there are mis-detection and missing detection of small traffic elements. At the same time, the difficulty of detection of juxtaposed dense traffic elements are also one of the problems restricting the update of traffic geographic information. This has caused some trouble to the automatic detection and recognition of road traffic elements. Therefore, it is important to improve the detection accuracy and precision of small traffic elements and juxtaposed dense traffic elements for the automatic detection and recognition of road traffic elements.

Small object detection is one of the most challenging problems in computer vision. Taking the definition of an object in COCO datasets [11] as an example, small objects refer

to objects with fewer than $32 \times 32$ pixel points. Small objects are easy to ignore in detection due to the small number of pixels in the image and the few available features, resulting in the phenomenon problem of missed detection. However, many scholars have improved the detection of small objects through strategies such as data enhancement, multiscale learning, contextual feature learning, and generative adversarial learning. For example, Yu et al. [12] proposed a vehicle detection method for aerial vehicles based on a deep neural network and traditional method. This method uses a deep segmentation network to mine the symbiotic relationship between roads and vehicles and then detects small vehicles based on a visual attention mechanism of spatiotemporal constraint information. Xiao et al. [13] proposed a semi-supervised fully convolutional neural network for extracting fine-grained road scene information using UAV images, which solved the problem of the high cost of labeled samples and implemented a fine-grained road scene resolution scheme for UAV remote sensing images. Wang et al. [14] proposed a multitask generative adversarial small object detection network which introduced artificial texture loss and center mask into the generator, making it easier for the generator to generate super resolution images and thus easier for small object detection. Xu et al. [15] proposed a feature enhancement network named FE-YOLO for remote sensing object detection to achieve real-time detection. FE-YOLO can accommodate remote sensing object detection under different backgrounds and can effectively improve the accuracy of remote sensing small object detection. Qing et al. [16] proposed a RepVGG-YOLO network that can be used for remote sensing image detection from arbitrary angles. Hu et al. [17] proposed a PAG-YOLO network by considering the global and local relationships in the input image. This network can adaptively redistribute the weight distribution of different scale features from spatial dimension and channel dimension through an attention-guided feature optimization module for the detection of small vessels. Kim et al. [18] proposed an ECAP-YOLO network, the model in which was based on the YOLOv5 network to discover small objects by adding a detection layer, reducing the computational power used for small target detection, improving its detection rate, and optimizing the detection of small objects. Liu et al. [19] proposed a method based on YOLOv3 for UAV perspective target detection. The method improves the whole network structure based on YOLOv3 by optimizing the resblock and adding convolution operations to enrich the spatial information and expand the UAV reception range for small target detection. Because the size of small objects only accounts for a small part of an image, their features are susceptible to the influence of weather illumination and occlusion, resulting in the phenomenon of detection error. Therefore, the detection of small objects is more difficult and challenging.

With the development of UAV technology, UAVs have been applied in all walks of life, such as the military, agriculture, electricity inspection, and transportation fields. UAVs play an important role in the transportation field because of their high efficiency, clear image acquisition, and high flexibility. UAVs can be used for cruising, vehicle tracking and identification, road traffic condition detection, etc. Lee et al. [20] proposed a set of comprehensive road monitoring systems using UAVs, aerial mapping cameras, and deep learning algorithms. The system can be used for road maintenance and management as well as autonomous vehicle roadmap planning. Some scholars use images collected by UAVs to monitor road damage, which greatly reduces the cost of manual visual interpretation. For example, Hong et al. [21] used UAVs to collect images of roads to detect and identify pavement cracks using an improved recognition method based on U-Net. The algorithm was made possible by a convolution block attention module, an improved encoder, and a strategy of fusing long and short skip connections. It enables the ability to predict road cracks with high accuracy on UAV images. Scholars have also implemented the extraction of road regions from UAV images. For example, Sultonov et al. [22] proposed an improved algorithm for automatic road extraction from UAV images, which is based on a lightweight model with depth separable convolution and ConvMixer initial blocks for extracting road regions from UAV images.

The main research of UAVs in the field of traffic and some strategies and methods for improving small target detection were introduced above. Similar to the above scholars, for this study, which collected road traffic element information by UAV, the problem that needs to be solved is the mis-detection and missed detection of small and juxtaposed dense road traffic elements. In our previous studies [1], we initially realized the automatic detection and recognition of road traffic element information. However, in the study, we also found that there are problems of incorrect and missed detection when detecting small and juxtaposed dense road traffic elements. Therefore, to solve the problems of poor detection of multiscale road traffic elements and difficult detection of small road traffic elements, an ASFF-YOLOv5 network is proposed for the automatic detection and recognition of road traffic multielements. The network improves the detection accuracy, reduces the problems of small object incorrect and missed detection, and realizes the automatic detection and recognition of road traffic multielements. It provides a new method for updating the basic traffic geographical information databases.

The main contributions of this paper are as follows:

(1)　The K-means++ [23] clustering method was used for data processing to obtain the optimal candidate box size of the object so that the detection anchor box is more consistent with a multielement road traffic dataset.

(2)　To address the problems of low detection accuracy, serious error detection and missed detections of road elements, and difficult recognition of small dense objects, an ASFF-YOLOV5 algorithm was proposed. In this algorithm, the classification accuracy and speed of multiple elements of road traffic are improved using the SPPF [24] structure. Moreover, by integrating the ASFF [25] structure of the RFB module [26], the receptive field is improved and the feature information of detection objects at different scales is improved to achieve richer feature information extraction, especially the detection and recognition ability of small objects.

(3)　The proposed ASFF-YOLOv5 algorithm was proven to be superior in detecting multiple elements of road traffic through comparative experiments and ablation experiments and provides a new solution for updating basic traffic geographical information databases.

The rest of this paper is organized as follows. Section 2 describes the datasets used in this study. Section 3 details the proposed method, followed by the experiments and results in Section 4. A discussion is presented in Section 5. Finally, our conclusions are outlined in Section 6.

## 2. Datasets and Scale Statistics

In this study, the multielement road traffic datasets produced in previous studies [1] were used as the experimental data, and the sample data are shown in Figure 1. The experimental data were captured by Hava MEGA-V8 and DJI FC6310 with a spatial resolution of 0.05 m and 0.1 m, respectively. In this study, representative multielement road traffic data were selected as the research object. The selected elements are zebra crossings, roadside parking spaces, and bus stations named zebra_crossing, parking_space, and bus_station. In the dataset division, the training data accounted for 90%, and the rest were the test data.

The default size of the anchor box in the YOLOv5 network is obtained by clustering according to the COCO dataset using the K-means algorithm [27], which is not applicable to the road traffic multielement datasets proposed in this paper. Therefore, to be more suitable for the object scale range of multielement road traffic datasets, the K-means++ clustering method was used to conduct scale statistics on UAV multielement road traffic remote sensing images in this study.

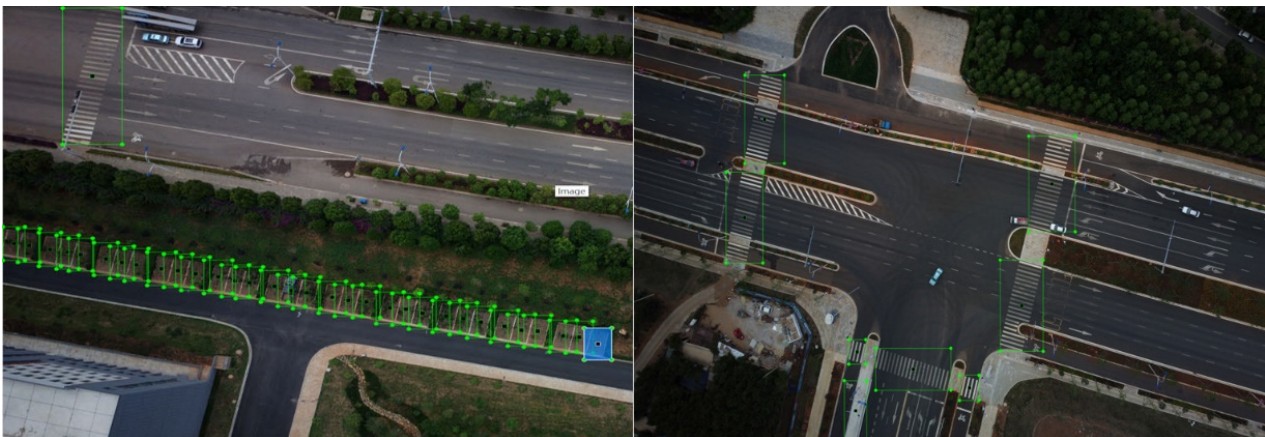

**Figure 1.** Sample dataset.

The K-means clustering algorithm is a typical iterative solution clustering algorithm. The core idea of the K-means clustering algorithm is to divide an object set into *n* clusters and randomly select a point as the cluster center. Iteration of the cluster center of each cluster continues until all the points in each cluster no longer change. However, the K-means clustering algorithm has some defects, and convergence depends heavily on the initialization of the cluster center. Compared with the K-means clustering algorithm, the K-means++ [23] clustering algorithm is improved in the initial random selection of the cluster center. Therefore, the K-means++ clustering algorithm was selected in this research for calculation. The K-means++ clustering algorithm selects cluster centers one by one instead of randomly when initializing the cluster centers, and the sample points farther away from other cluster centers are more likely to be selected as the next cluster center. The initial cluster center was obtained using the K-means++ cluster center calculation, and then the clustering result was obtained by the K-means clustering algorithm. In this study, the road traffic element scale was defined as nine clusters, and the object candidate box size was calculated by the K-means++ clustering algorithm. The results after K-means++ clustering from the test results of different K-means in Section 4.3 are shown in Table 1. Compared with the nonuse of the K-means++ algorithm, the mean average accuracy increase was 18.3%. Compared with K-means++ clustering, the mean average accuracy increase was 3%. The result shows that the effect of K-means++ clustering conformed to the datasets of road traffic elements.

**Table 1.** The results of K-means++ clustering.

| Serial No. | 1 | 2 | 3 | 4 | 5 | 6 | 7 | 8 | 9 |
|---|---|---|---|---|---|---|---|---|---|
| $x$ | 10 | 15 | 16 | 19 | 23 | 25 | 33 | 52 | 125 |
| $y$ | 17 | 29 | 19 | 38 | 22 | 26 | 70 | 34 | 123 |

## 3. Research Method

The proposed multielement road traffic detection method based on multiscale feature fusion includes data preprocessing, feature extraction, and feature fusion. First, the K-means++ clustering algorithm is used for preprocessing, and the size of the candidate box matching the multielement datasets of road traffic is obtained. Then, the main feature extraction network integrated with an SPPF structure is used to extract the features of the input feature map to achieve speed improvement and complete richer feature information extraction. Second, through a fusion of an ASFF module after the feature extraction is completely strengthened, the features of the pyramid through an RFB module expand the receptive field. To further extract the features of different scale road traffic elements, the features of the pyramid in the implementation of multiscale road traffic element feature fusion makes full use of the features of road traffic elements of different maps and the

details of the semantic information. Finally, the detection results of multielement road traffic are output through the detection head, and their accuracy is evaluated. The specific flow of the proposed method is shown in Figure 2.

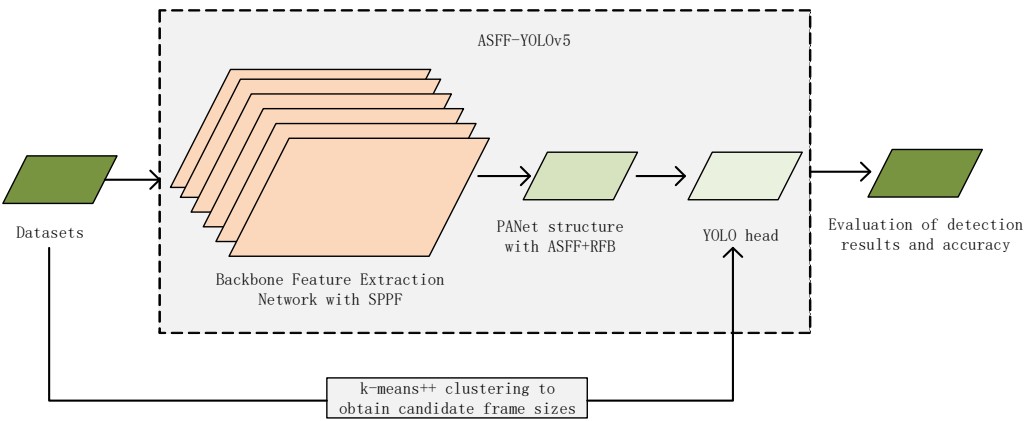

**Figure 2.** Flow chart of the proposed method.

### 3.1. ASFF-YOLOv5

The pyramid module in the original YOLOv5 network [24] was borrowed from PANet [28], and the YOLOv5 network structure diagram is shown in Figure 3. PANet achieves multiscale feature fusion of the network by aggregating features from different backbone layers and between detection layers. However, in PANet feature fusion, feature information is extracted from deep and shallow feature maps continuously, and then the feature map scale is transformed into the same scale. Then, simple addition is performed to obtain feature fusion, which cannot make full use of feature information of different scales. Therefore, the proposed ASFF-YOLOv5 algorithm makes full use of the feature information of different scales to improve the detection effect of different scale objects.

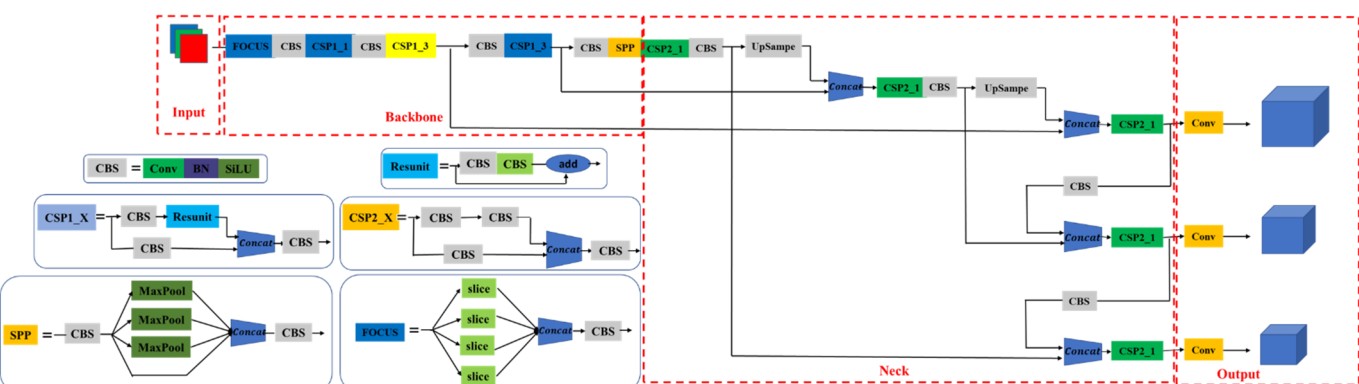

**Figure 3.** Network structure diagram of YOLOv5.

The ASFF-YOLOv5 network consists of three parts: a backbone feature extraction network, a feature map pyramid network, and a classifier and regressor. In the ASFF-YOLOv5 network, the backbone feature extraction network is used to extract the input feature map. The feature map pyramid network achieves the feature fusion of multiscale road traffic elements. The classifier and regressor obtain the detection results. Figure 4 shows a schematic diagram of the proposed method, in which the deepened color part is the proposed SPPF module and ASFF + RFB module. Assuming that the size of the input feature map in the network is $640 \times 640 \times 3$, the feature extraction process of the ASFF-YOLOv5 network is as follows:

(1)  The height and width are compressed in the feature layer by focusing the structure, and the number of expansion channels is quadrupled to obtain a $320 \times 320 \times 12$ feature map.

(2)  A $320 \times 320 \times 64$ feature map is obtained through a series of operations, such as convolution, normalization, and activation functions.

(3)  In the backbone feature extraction network, three effective feature layers are obtained by stacking residual extraction, and the SPPF structure is introduced in the last effective feature layer. The SPPF structure improves the classification accuracy and speed of the feature map by using the maximum pooling of the same pooling kernel for feature extraction; at this time, the three effective feature layers obtained in the backbone feature extraction network are $80 \times 80 \times 256$, $40 \times 40 \times 512$, and $20 \times 20 \times 1024$.

(4)  The obtained effective feature layer is transferred to the PANet structure, and the feature extraction is further enhanced through upsampling and downsampling. In this stage, the proposed ASFF + RFB module is integrated and used for the extraction of multiscale road traffic element information. It can achieve the enhancement of the perceptual field, realize the extraction of feature information of detectors at different scales, and complete the extraction of richer feature information.

(5)  Three enhanced effective feature layers are obtained, and the prediction and regression results are obtained by the classifier and regressor.

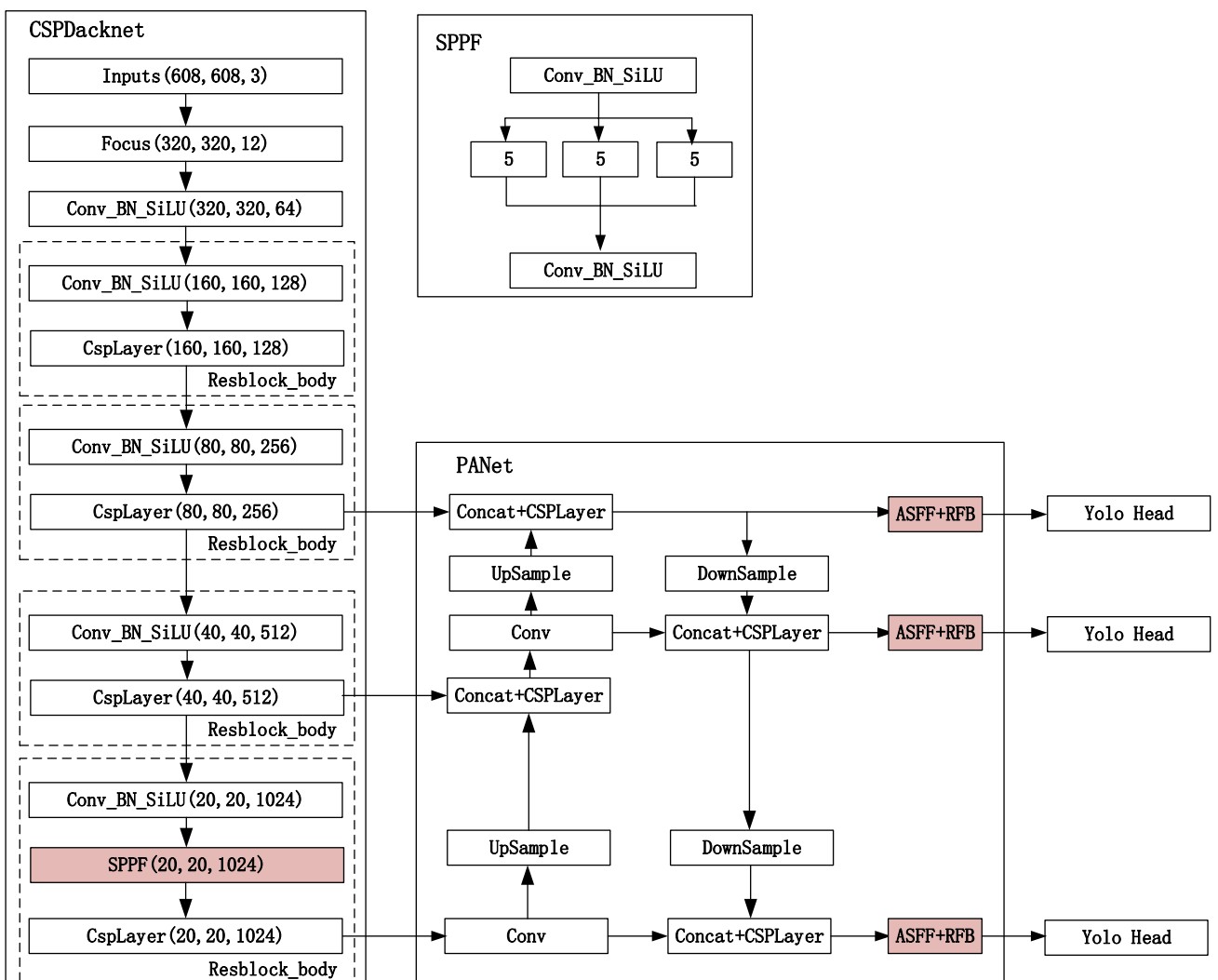

**Figure 4.** ASFF-YOLOv5 structure diagram.

*3.2. ASFF + RFB Module*

The ASFF-YOLOv5 algorithm enables improved feature scale invariance and object detection by fusing the ASFF structure into the PANet structure. In the PANet structure of the ASFF-YOLOv5 algorithm, the ASFF algorithm is introduced in each layer of the FPN structure for weighted fusion after first enhancing the semantic feature extraction top-down for the FPN structure [29]. The weight parameters are derived from the output of the convolutional feature layer, and the weight parameters become learnable after gradient backpropagation so that they can be adaptive when performing weighted fusion. Meanwhile, an RFB module layer is introduced after the ASFF algorithm. This module fuses the null convolution on top of Inception [30], thus effectively increasing the perceptual field and realizing the feature extraction capability of the network. The PANet structure of the proposed fused ASFF algorithm is shown in Figure 5.

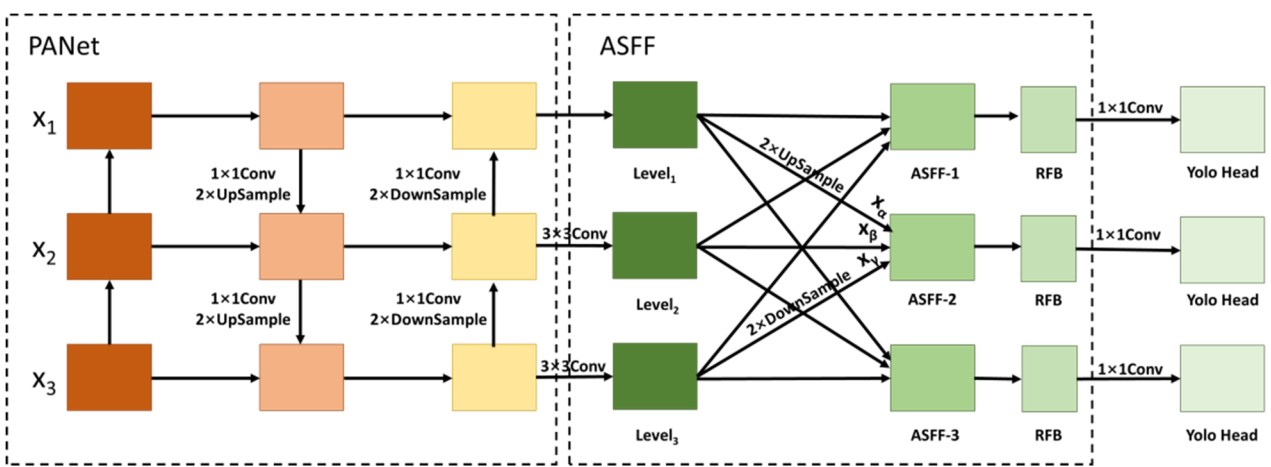

**Figure 5.** Integration of the PANet structure into the ASFF module.

Taking ASFF-2 computational fusion as an example, $X_1$, $X_2$, and $X_3$ are the feature maps extracted from the YOLOv5 backbone network. First, the feature maps $Level_1$, $Level_2$, and $Level_3$ are obtained from the PANet structure. Then, ASFF-2 is obtained by fusing them with the ASFF algorithm. The $Level_1$ feature map is convolved to obtain the same number of channels as the $Level_2$ feature map, and then the feature map with the same dimension as $Level_2$ is upsampled to obtain $X^{1\to2}$. For the $Level_3$ feature map, the number of channels and dimensions are adjusted using convolution and downsampling operations to keep the same number of channels and dimensions as $Level_2$, and $X^{3\to2}$ is obtained. The $Level_2$ feature map is adjusted by the number of channels after the convolution operation to obtain $X^{2\to2}$. After processing the three feature maps using the softmax function, the weight coefficients $\alpha$, $\beta$, and $\gamma$ of $X^{1\to2}$, $X^{2\to2}$ and $X^{3\to2}$ are obtained, respectively, and then the ASFF fusion calculation is performed with the following formula.

$$y_{ij}^l = \alpha_{ij}^l \cdot X_{ij}^{1\to l} + \beta_{ij}^l \cdot X_{ij}^{2\to l} + \gamma_{ij}^l \cdot X_{ij}^{3\to l} \tag{1}$$

where $y_{ij}^l$ is the new feature map obtained using the ASFF module. $\alpha_{ij}^l$, $\beta_{ij}^l$, and $\gamma_{ij}^l$ are the weight coefficients of the three feature maps, and $\alpha_{ij}^l$, $\beta_{ij}^l$, and $\gamma_{ij}^l$ after softmax function processing satisfy $\alpha_{ij}^l + \beta_{ij}^l + \gamma_{ij}^l = 1$, $\alpha_{ij}^l$, $\beta_{ij}^l$, and $\gamma_{ij}^l \in [0,1]$. $X_{ij}^{n\to l}$ denotes the feature vector of the feature map from layer $n$ to layer $l$.

Through the ASFF algorithm, the multiscale feature fusion of the model is fully realized by adjusting the feature fusion with the weight parameters. In addition, this study introduces the RFB module along with the ASFF algorithm. Through multiple branch convolution and dilated convolution, the RFB module can increase the receptive field more effectively, improve the utilization of feature information, and improve the model's ability to recognize and detect small objects. In the multiple branch structures of the RFB module,

the first layer of each branch is composed of convolution kernels of a specific size, and the size of the first layer convolution kernels is 1 × 1, 3 × 3, and 5 × 5. Figure 6a shows the effect diagram of the RFB module, and Figure 6b shows the structure diagram of the RFB module. The rate represents the expansion coefficients of different dilated convolution layers. The RFB module includes a dilated convolution layer to enhance the receptive field. The final output of the RFB module concatenates the output feature maps of different sizes and receptive fields to achieve the purpose of fusing different features.

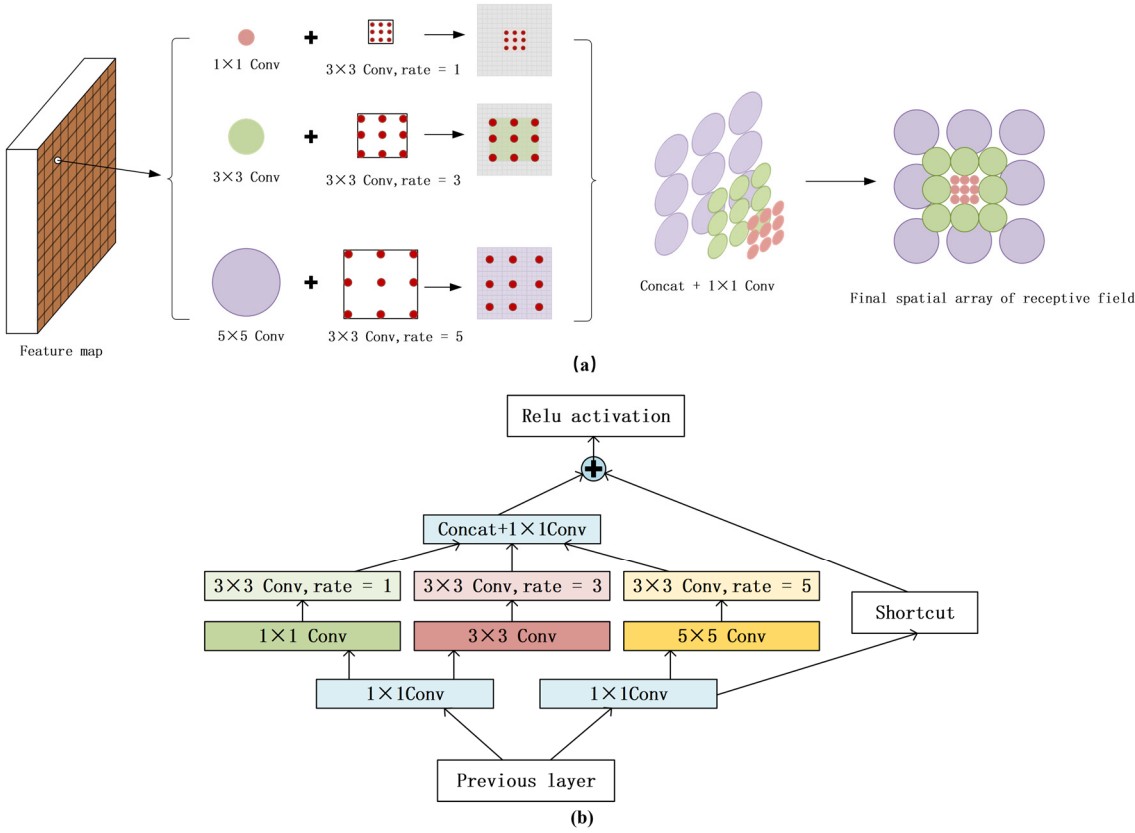

**Figure 6.** Schematic diagrams of (**a**) the RFB module effect and (**b**) the RFB structure.

### 3.3. Spatial Pyramid Pooling Fast

The role of the SPP structure [31] in the YOLOv5 network is to implement a fixed size feature vector as a fully connected layer output for images with different size inputs. The SPP structure uses three convolution kernels of different sizes, 3, 5, and 9, to extract features through the maximum pooling operation, enhance the feature expression ability of the feature graph, and improve the network receptive field. The SPP structure is shown in Figure 7a, which first performs 1 × 1, 3 × 3, 5 × 5, and 9 × 9 maximum pooling operations on the data transferred from the convolutional normalization activation function (Convolution + Banch Normalization + SiLU, CBS) in parallel and then connects them to the CBS structure by concatenation splicing. The CBS structure achieves feature fusion and completes the feature extraction operation. However, the SPP structure increases the computation of the program by parallel pooling operations with different sized convolutional kernels, which degrade the performance. Therefore, the SPPF structure is used for pooling to improve the performance of pooling while reducing the amount of program computation. The SPPF structure replaces the parallel maximum pooling operation of three convolutional kernels of different sizes in the original SPP with a serial operation of three convolutional kernels of the same size. As shown in Figure 7b, the SPPF structure is similar to the SPP structure. The operation of the SPPF layer first performs a 5 × 5 maximum pooling operation on the data serially transferred from the CBS structure. Then, the data

are passed into the CBS structure by concatenation splicing, which achieves a speed-up while completing a richer feature information extraction.

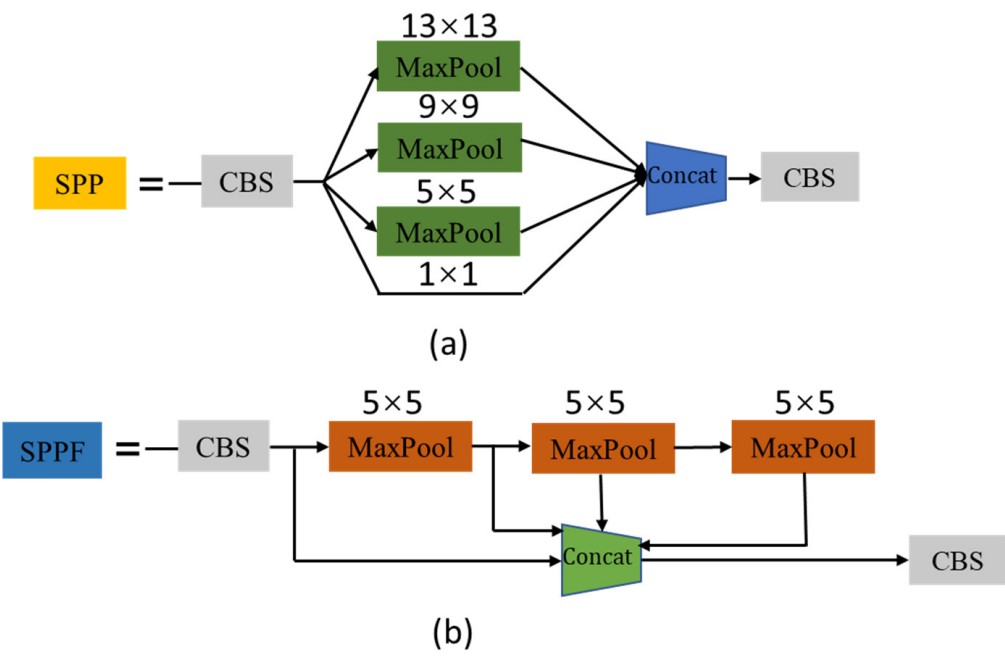

**Figure 7.** (**a**) SPP structure diagram and (**b**) SPPF structure diagram.

*3.4. EIoU Loss*

Through the above method, the object detection accuracy and overall recognition efficiency of multielement road traffic are improved. However, the overall detection accuracy is reduced if there is a parallel and dense phenomenon in road traffic elements, such as roadside parking spaces. Because the anchor boxes of the parallel and dense object detection overlap and block each other, part of the overlap is removed in the nonmaximal suppression process, leading to the phenomenon of missed detection. In previous studies [1], the CIoU loss function [32] was used to solve this problem. The CIoU loss function calculation considered the location information between the object box and the detection box, such as the aspect ratio of the distance overlap area. When there is no overlap between the object box and the detection box, backpropagation can still be carried out. However, in the process of prediction box regression, if the prediction box and the true box increase or decrease proportionally, the penalty term [32] defined by the CIoU loss function no longer takes effect. Therefore, the EIoU loss function [33] was adopted in this study to solve this problem. Compared with the CIoU loss function, EIoU [33] directly calculates penalty terms for the width and height, avoiding the problem that the penalty terms do not work when the prediction box and true box increase or decrease proportionally. At the same time, the regression accuracy is improved using the EIoU so that more attention is given to high-quality anchor boxes in the regression process.

The EIoU loss consists of three components, which are the overlap loss of the predicted and true boxes, the center distance loss of the predicted and true boxes, and the edge length loss of the height and width of the predicted and true boxes. Therefore, the EIoU loss function is defined as:

$$EIoU_{Loss} = 1 - IoU + \frac{\rho^2(b, b^{gt})}{c^2} + \frac{\rho^2(w, w^{gt})}{C_w^2} + \frac{\rho^2(h, h^{gt})}{C_h^2} \qquad (2)$$

where $IoU$ [34] is the intersection ratio of the area of the prediction box and the real box. $\rho^2(b, b^{gt})$ denotes the Euclidean distance between the prediction box and the center point of the true box. $\rho^2(w, w^{gt})$ denotes the Euclidean distance between the predicted box and the true box width. $\rho^2(h, h^{gt})$ denotes the Euclidean distance between the predicted box

and the true box height. $c$ denotes the diagonal distance between the prediction box and the minimum outer rectangle of the true box. $C_w$ denotes the closure width between the prediction box and the true box. $C_h$ denotes the closure height between the prediction box and the true box.

## 4. Experimental Results and Analysis

### 4.1. Experimental Environment

The computer hardware configuration used in this study is an Ubuntu 20.04 system with an Intel i7-8700 CPU and a GTX1070 graphics card configuration with 8 GB video memory. The training was performed using PyCharm software version 2020.1 (downloadable from https://www.jetbrains.com/pycharm/download/ (accessed on 1 November 2021), Prague, Czech Republic). In the experimental training, the weight decay coefficient of training was set to 0.0005, the initial learning rate was 0.001, the confidence level was 0.5, the IoU threshold was set to 0.65, and a total of 100 epochs were trained with 4000 iterations.

### 4.2. Evaluation Indicators

The experiments measured the accuracy of model detection by calculating the *mAP*, average precision (*AP*), precision, and recall as model quantitative evaluation metrics, which were defined as shown in Table 2.

**Table 2.** Evaluation indicators.

| Evaluation Indicators | Calculation Formula | Definition |
|---|---|---|
| *Precision* | $Precision = \frac{TP}{TP+FP}$ | *TP* represents the positive samples detected correctly, indicating the number of road traffic elements detected correctly. *FP* represents the negative samples detected incorrectly, indicating the number of objects detected as road traffic element classes but actually other classes. *FN* represents the positive samples detected incorrectly, indicating the number of other classes detected as actually road traffic element classes. |
| *Recall* | $Recall = \frac{TP}{TP+FN}$ | |
| *AP* | $AP = \int_0^1 p(r)d_r$ | The value of *AP* is the size of the area enclosed by $p$ as a function of $r$ in the range [0, 1], where $p$ is the precision and $r$ is the recall. |
| *mAP* | $mAP = \frac{\sum_{i=1}^{N} AP_i}{N}$ | *N* represents the number of all categories in the test set, $i$ is the $i$th category, and $AP_i$ is the average precision rate of the $i$th category. |

### 4.3. Comparison Experiments

To verify the effectiveness of the proposed method, the classical algorithm network of object detection was compared in this study. The SSD [35], Retinanet [36], Faster R-CNN [37], YOLOv3 [38], YOLOv4 [39], YOLOv5 [24] networks as well as the previous research were selected for comparison experiments with the proposed method. In the experiment, the values of the *AP*, precision, recall, and *mAP* evaluation indexes were calculated and compared when the multielement datasets of road traffic were trained. As shown in Table 3, the recognition accuracy of multielement road traffic under different network models was counted. Among them, the increased *mAP* values were calculated by comparing each network and the proposed method.

**Table 3.** Detection results of different networks.

| Network Model | Transport Elements | AP/% | Precision/% | Recall/% | mAP/% | Rise Points |
|---|---|---|---|---|---|---|
| Faster R-CNN | zebra crossings | 64.3 | 59.0 | 71.4 | 56.9 | 36.2 |
| | bus stations | 71.5 | 73.3 | 71.0 | | |
| | roadside parking spaces | 35.0 | 31.5 | 48.8 | | |
| Retinanet | zebra crossings | 70.0 | 87.8 | 61.2 | 57.3 | 35.8 |
| | bus stations | 67.3 | 86.4 | 61.3 | | |
| | roadside parking spaces | 34.5 | 74.3 | 24.4 | | |
| SSD | zebra crossings | 52.9 | 76.8 | 32.6 | 53.9 | 39.2 |
| | bus stations | 75.1 | **100.0** | 54.8 | | |
| | roadside parking spaces | 33.8 | 73.4 | 14.7 | | |
| YOLOv3 | zebra crossings | 84.3 | 87.4 | 80.4 | 81.5 | 11.6 |
| | bus stations | 83.8 | 88.9 | 77.4 | | |
| | roadside parking spaces | 76.5 | 76.1 | 75.9 | | |
| YOLOv4 | zebra crossings | 81.8 | 90.0 | 79.2 | 74.7 | 18.4 |
| | bus stations | 76.8 | 90.5 | 61.3 | | |
| | roadside parking spaces | 65.4 | 70.2 | 71.0 | | |
| YOLOv5 | zebra crossings | 85.3 | 75.2 | 82.0 | 73.9 | 19.2 |
| | bus stations | 65.2 | 50.9 | 58.1 | | |
| | roadside parking spaces | <u>78.9</u> | 74.2 | 77.5 | | |
| YOLOv4 + ECA | zebra crossings | **94.3** | **90.1** | <u>93.9</u> | <u>90.5</u> | 2.6 |
| | bus stations | **99.6** | 91.3 | **100.0** | | |
| | roadside parking spaces | 77.4 | <u>81.0</u> | <u>78.1</u> | | |
| ASFF-YOLOv5 (YOLOv5 + ASFF + RFB) | zebra crossings | <u>94.0</u> | 85.9 | 94.4 | 93.1 | |
| | bus stations | <u>96.2</u> | <u>93.1</u> | <u>96.3</u> | | |
| | roadside parking spaces | **83.7** | **86.9** | **88.6** | | |

Note: the best results are in bold font and the second best results are underlined.

To verify the accuracy of anchor positioning obtained by K-means++ clustering in this study, the experimental results of the original YOLOv5 network, YOLOv5 + K-means, and YOLOv5 + K-means++ were compared. Table 4 shows the detection results of different K-means.

**Table 4.** Detection results of different K-means.

| Network Model | Transport Elements | AP/% | Precision/% | Recall/% | mAP/% | Rise Points |
|---|---|---|---|---|---|---|
| YOLOv5 | zebra crossings | 85.3 | 75.2 | 82.0 | 73.9 | 18.3 |
| | bus stations | 65.2 | 50.9 | 58.1 | | |
| | roadside parking spaces | 78.9 | 74.2 | 77.5 | | |
| YOLOv5 + K-means | zebra crossings | <u>93.4</u> | <u>84.0</u> | <u>92.5</u> | <u>89.2</u> | 3 |
| | bus stations | <u>95.3</u> | <u>74.0</u> | <u>83.4</u> | | |
| | roadside parking spaces | <u>84.7</u> | <u>84.9</u> | <u>88.7</u> | | |
| YOLOv5 + K-means++ | zebra crossings | **94.6** | **84.8** | **93.8** | **92.2** | |
| | bus stations | **95.5** | **78.2** | **93.9** | | |
| | roadside parking spaces | **86.0** | **86.9** | **88.9** | | |

Note: the best results are in bold font and the second best results are underlined.

To verify the accuracy of the K-means++ clustering effect on anchor boxes, images were selected for detection, and the detection effect under different K-means is shown in Figure 8. From the viewpoint of detection accuracy and the size and location of the anchor box, YOLOv5 had the lowest detection accuracy for the detectors, with an accuracy of only 70.6%. The location of the candidate box does not include all the detectors to be detected. The detection accuracy of YOLOv5 + K-means for detected objects ranked second, with an accuracy of 87.9%. Although the positioning accuracy of the candidate boxes was improved, there was still a slight gap. YOLOv5 + K-means++ had the highest detection accuracy of 95.7% for detected objects, the positioning of candidate boxes was the most accurate, there was no missing part, and the size of candidate boxes was appropriate. The results showed that the K-means++ clustering algorithm used could make the position of candidate boxes more accurate and the detection effect better.

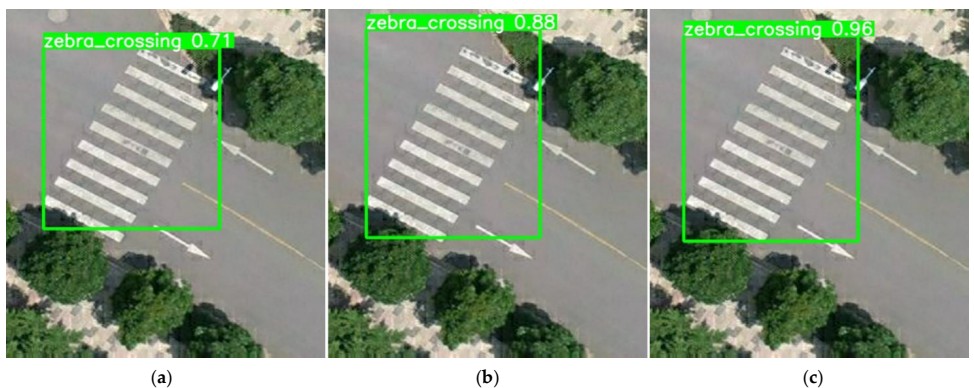

| (a) | (b) | (c) |

**Figure 8.** Detection effects under different K-means: (**a**) YOLOv5; (**b**)YOLOv5 + K-means; and (**c**) YOLOv5 + K-means++.

To verify the effectiveness of the proposed algorithm, an ablation experiment was conducted on the multielement road traffic datasets, and the experimental results after fusion of K-means++, SPPF, and ASFF were compared. The AP, precision, recall, and mAP values were calculated and compared. The ablation test results are shown in Table 5.

**Table 5.** Results of ablation tests.

| Network Model | Transport Elements | AP/% | Precision/% | Recall/% | mAP/% | Rise Points |
|---|---|---|---|---|---|---|
| YOLOv5 | zebra crossings | 85.3 | 75.2 | 82.0 | 73.9 | 19.2 |
| | bus stations | 65.2 | 50.9 | 58.1 | | |
| | roadside parking spaces | 78.9 | 74.2 | 77.5 | | |
| YOLOv5 + SPPF | zebra crossings | **94.9** | 90.3 | 94.2 | 90.7 | 2.4 |
| | bus stations | **100.0** | 83 | 88.8 | | |
| | roadside parking spaces | <u>85.6</u> | 86.2 | **89.1** | | |
| YOLOv5 + K-means++ | zebra crossings | <u>94.6</u> | 84.8 | 93.8 | 92.2 | 0.9 |
| | bus stations | 95.5 | 78.2 | 93.9 | | |
| | roadside parking spaces | **86.0** | 86.9 | 88.9 | | |
| YOLOv5 + ASFF | zebra crossings | 91.3 | **92.1** | 94.1 | 92.8 | 0.3 |
| | bus stations | **100.0** | <u>91.6</u> | <u>95.1</u> | | |
| | roadside parking spaces | 77.1 | **91.4** | <u>89.0</u> | | |
| ASFF-YOLOv5 (YOLOv5 + ASFF + RFB) | zebra crossings | 94.0 | 85.9 | **94.4** | **93.1** | |
| | bus stations | <u>96.2</u> | **93.1** | **96.3** | | |
| | roadside parking spaces | 83.7 | <u>86.9</u> | 88.6 | | |

Note: the best results are in bold font and the second best results are underlined.

To verify the practicality and effectiveness of the proposed method, prediction experiments were conducted in different scenes separately. Small scene image maps taken by an UAV and large scene image maps generated by commercial software processing were selected for detection. The small scene images were normal road scenes and small object element scenes. Among them, the normal road scenes were intersection sections imaged by the UAV, including 1 bus station and 10 zebra crossings. The small object element scenes were roadside parking spaces captured by the UAV, divided into scene 1 and scene 2. Scene 1 was an unobstructed roadside parking space, including nine juxtaposed and dense roadside parking spaces. Scene 2 was an obscured object to be detected, including 45 roadside parking spaces and 2 zebra crossings. The parking spaces were juxtaposed, dense, and partially obscured by trees. The prediction effect under the normal road scene is shown in Figure 9. Figure 10 shows scene 1 under small target elements. Figure 11 shows scene 2 under small target elements. Table 6 shows the detection results under small scenes.

The prediction effect can be seen from the results shown in Figure 9 and Table 6. In the normal road scenario, the ablation experiments were able to correctly detect the multielement road traffic with no detection error or missed detections. The ablation experiments all had good ability to detect multiple elements of road traffic. However, the detection accuracy obtained by the proposed method was the highest.

Figure 10 shows the unobstructed detection in the small target element scenario, where the object detection was nine side-by-side and dense roadside parking spaces. Figure 10 and Table 6 show that the ablation experiments detected all roadside parking spaces in the small object scenario without obstruction and there were no detection errors or missed detections. Ablation experiments have a good ability to detect multiple elements of road traffic in unobstructed environments. However, the detection accuracy of the method in this paper reached 91.2% when detecting small objects without occlusion, which is the highest compared with other methods.

Figure 11 shows the detection with occlusion in the small object element scene, and the detection objects were 45 parallel and dense roadside parking spaces and 2 zebra crossings. Figure 11 and Table 6 show that the ablation experiments achieved 99.5% and 100% detection of zebra crossings, respectively. It is proven that the above ablation experiments could achieve good results for zebra crossing detection and that there were no detection errors or missed detections. However, the above ablation experiments were missed in the detection of street parking spaces, and the number of missed street parking spaces ranged from 3 to 11. From the overall results, although the YOLOv5 + SPPF method resulted in the highest average detection accuracy, the number of missed spots was the most serious at 11. Compared with the serious missed detections in the ablation experiment, the proposed method only missed three detections, which was the minimum number of missed detections in the ablation experiment. The detection accuracy of the proposed method was 93.5% for obscured roadside parking spaces, which is only different by 0.28 from the optimal detection accuracy, but the missed detections were greatly reduced. This proves that the proposed method ensured the correctness of detection while maintaining the detection accuracy, especially in the detection of obscured small objects and juxtaposed dense small objects with high improvement.

Figure 12 shows the prediction result map under an orthophoto image. The image is an orthophoto map generated by a remote sensing image map taken by UAV and processed using commercial software, and the total area of this image area is 302,813 m$^2$. The image size is 37,548 × 21,438 pixels. The small and medium objects in this image are 140 × 82 pixels and 514 × 254 pixels, respectively. The image contains 18 zebra crossings, 5 bus stations, and 58 roadside parking spaces. The specific prediction results are shown in Table 7.

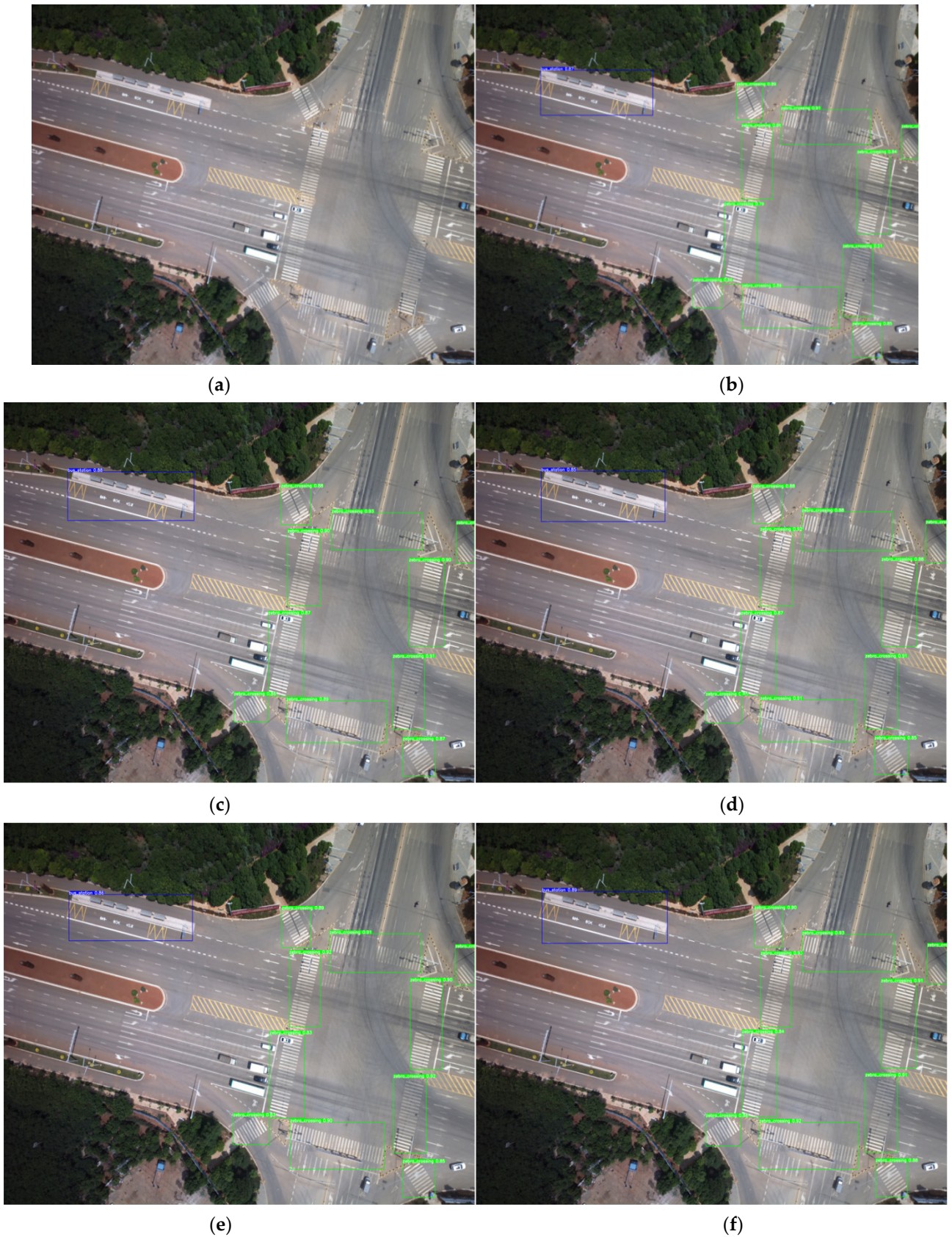

**Figure 9.** Detection results in normal road scenarios: (**a**) detection of the original image; (**b**)YOLOv5;
(**c**) YOLOv5 with SPPF; (**d**) YOLOv5 with K-means++; (**e**) YOLOv5 with ASFF; and (**f**) ASFF-YOLOv5.

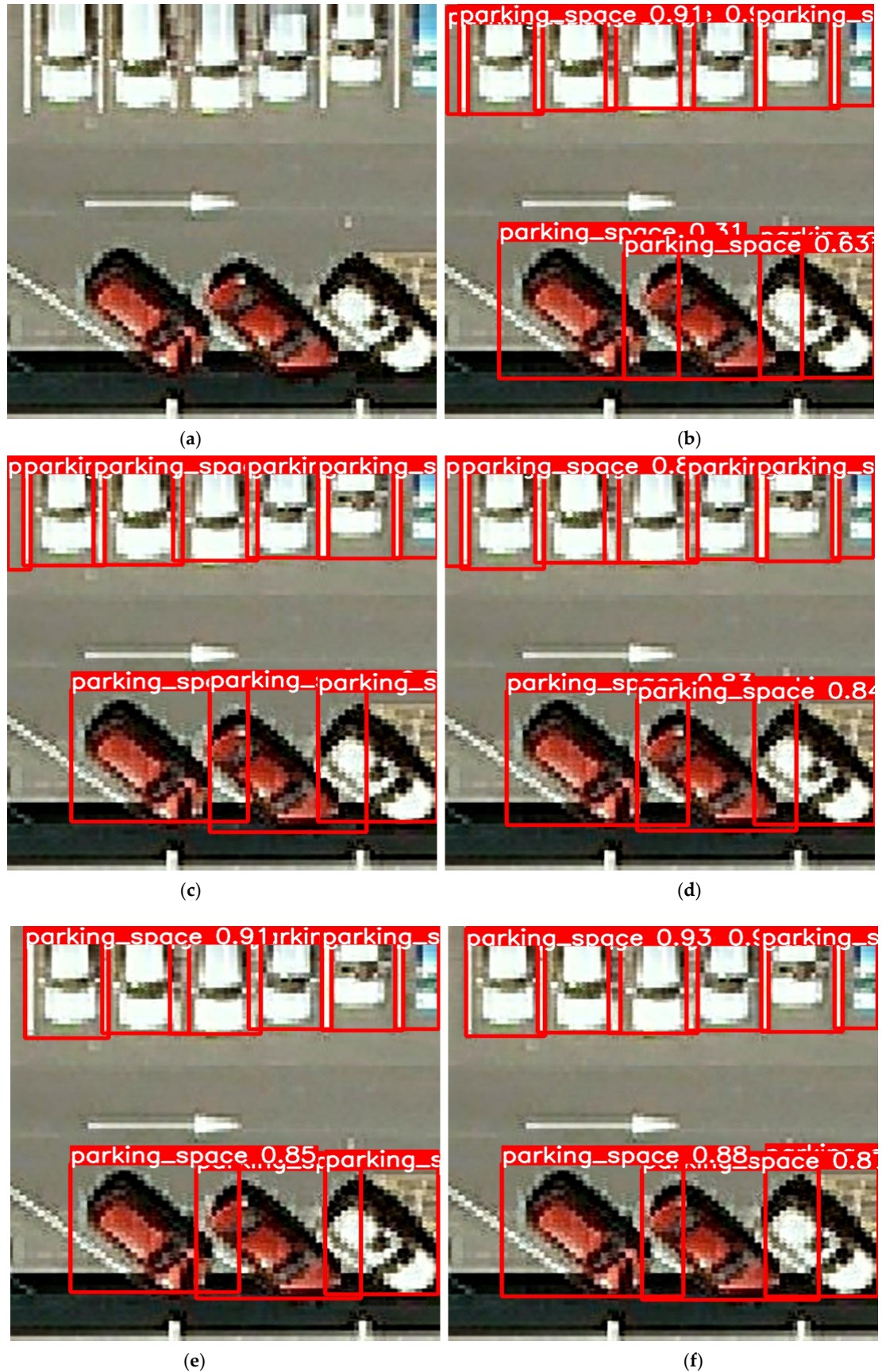

**Figure 10.** Unobstructed detection in small object scenes: (**a**) detection of the original image; (**b**)YOLOv5; (**c**) YOLOv5 with SPPF; (**d**) YOLOv5 with K-means++; (**e**) YOLOv5 with ASFF; and (**f**) ASFF-YOLOv5.

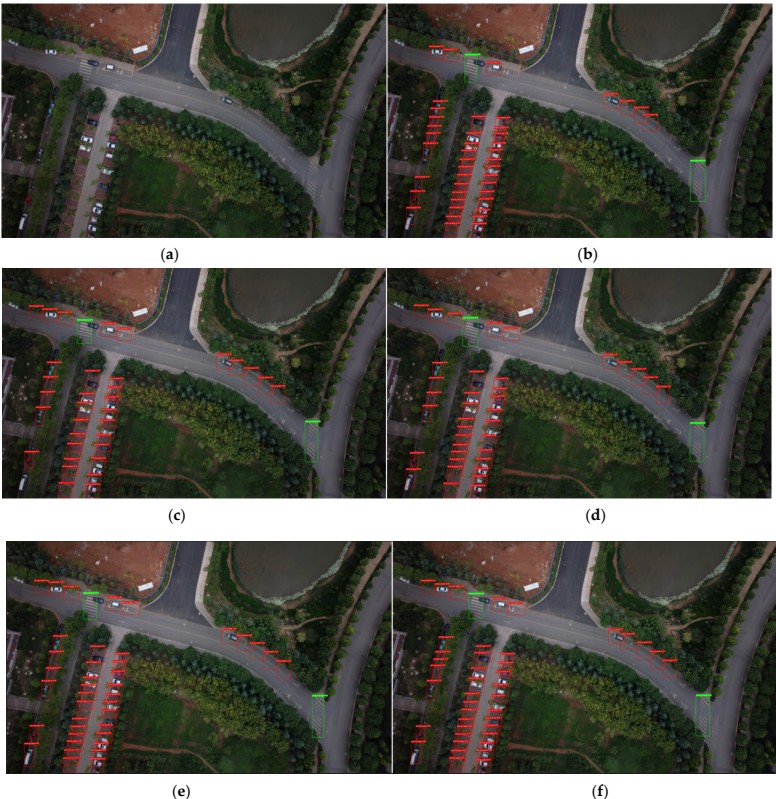

**Figure 11.** Effect of detection with occlusion in small target scenes: (**a**) detection of the original image; (**b**) YOLOv5; (**c**) YOLOv5 with SPPF; (**d**) YOLOv5 with K-means++; (**e**) YOLOv5 with ASFF; and (**f**) ASFF-YOLOv5.

**Table 6.** Predicted results of ablation experiments in a small scenario.

| Network Model | Transport Elements | Normal Road Scene | | Small Target Scenes Scene 1 (Unobstructed) | | Small Target Scenes Scene 2 (Obstructed) | |
|---|---|---|---|---|---|---|---|
| | | Number | AP/% | Number | AP/% | Number | AP/% |
| YOLOv5 | zebra crossings | 10 | 87.0 | - | - | 2 | 87 |
| | bus stations | 1 | 81.7 | - | - | - | - |
| | roadside parking spaces | - | - | 9 | 72.9 | 42 | 81.6 |
| YOLOv5 + SPPF | zebra crossings | 10 | 85.3 | - | - | 2 | 99.5 |
| | bus stations | 1 | 87.8 | - | - | - | - |
| | roadside parking spaces | - | - | 9 | 76.6 | 34 | **96.3** |
| YOLOv5 + K-means++ | zebra crossings | 10 | 87.1 | - | - | 2 | **100** |
| | bus stations | 1 | 84.8 | - | - | - | - |
| | roadside parking spaces | - | - | 9 | 82.9 | 38 | 93.3 |
| YOLOv5 + ASFF | zebra crossings | 10 | 86.7 | - | - | 2 | **100** |
| | bus stations | 1 | 85.7 | - | - | - | - |
| | roadside parking spaces | - | - | 9 | 89.7 | 40 | 93.3 |
| ASFF-YOLOv5 (YOLOv5 + ASFF + RFB) | zebra crossings | 10 | **88.4** | - | - | 2 | 99.5 |
| | bus stations | 1 | **89.1** | - | - | - | - |
| | roadside parking spaces | - | - | 9 | **91.2** | 42 | 93.5 |

Note: "-" in the table indicates that the images measured do not contain this category, and the bolded font is the best result for each. Number is the number of road traffic elements that were correctly detected.

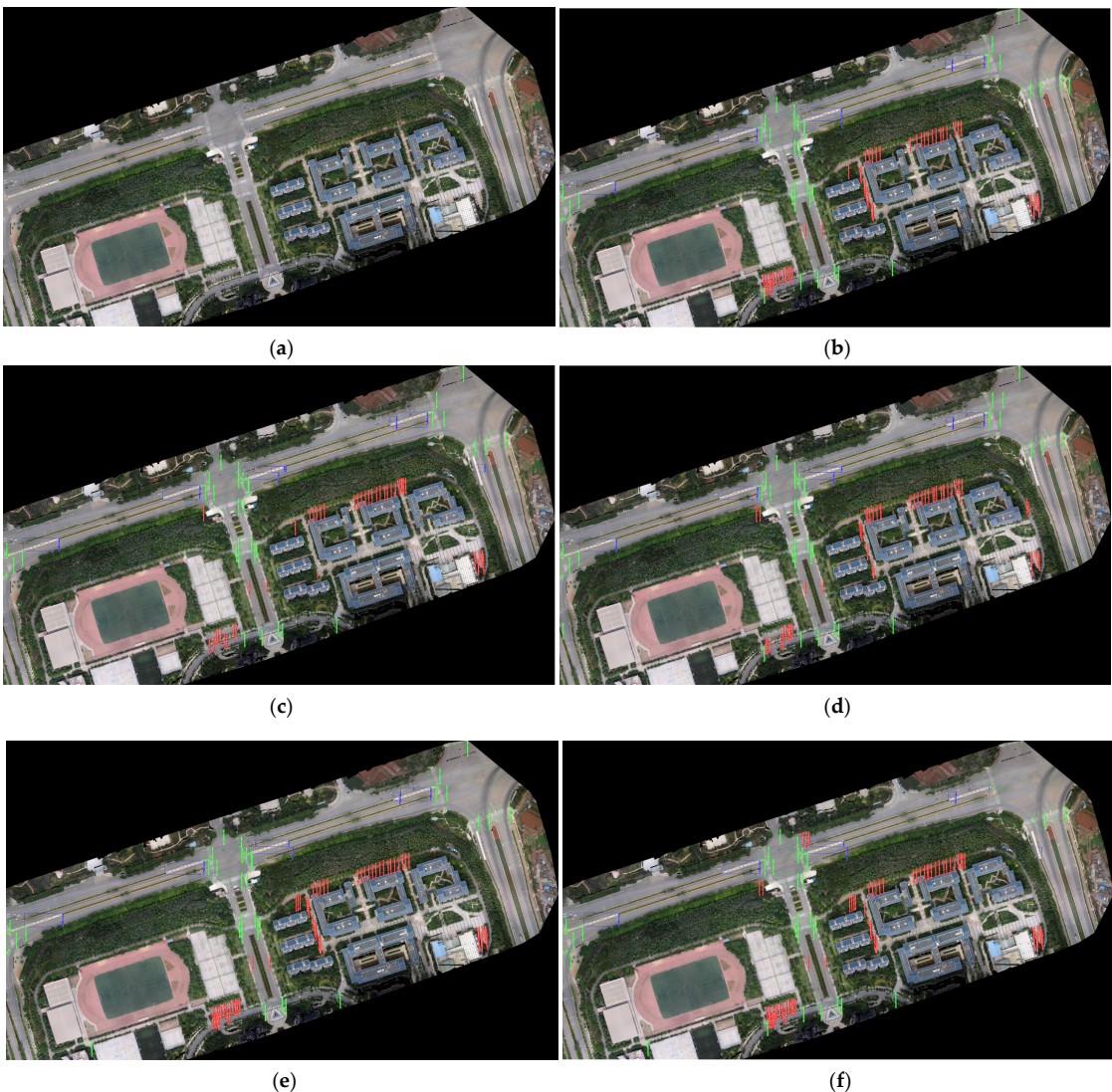

**Figure 12.** Predicted results of the ablation experiment for a large and complex scenario: (**a**) detection of the original image; (**b**) YOLOv5; (**c**) YOLOv5 with SPPF; (**d**) YOLOv5 with K-means++; (**e**) YOLOv5 with ASFF; and (**f**) ASFF-YOLOv5.

Figure 12 and Table 7 show that several of the above algorithms showed different degrees of missed detections when applied to complex large scenes. However, the proposed method had the lowest number of missed detections. When detecting bus stations and zebra crossings, the missed detections in the ablation experiment were not obvious, and all methods could correctly detect the objects. However, in the detection of roadside parking spaces, the missed detections were serious, and the number of missed detections ranged from 8 to 29. The reason for this is that for large images, roadside parking is a small target detection and most of the parking spaces are covered by greenery, so the feature information is not obvious, thus causing missed detections. The proposed method is optimized for multiscale feature extraction, which improved the detection accuracy of small objects to a certain extent. The proposed method achieved an average detection accuracy of 80.3% for roadside parking spaces in complex and large scenes. Compared with the ablation experiments, the number of missed detections was the lowest, and only eight roadside parking spaces obscured by greenery were missed. The average detection accuracy for zebra crossings and bus stations in complex large scenes reached 89.5% and 89.7%, respectively, which were the best results of the ablation experiment.

**Table 7.** Predicted results of ablation experiments for a large and complex scenario.

| Network Model | Transport Elements | Number 1 | Number 2 | AP/% |
|---|---|---|---|---|
| YOLOv5 | zebra crossings | 18 | 0 | 87.2 |
| | bus stations | 4 | 1 | 75.5 |
| | roadside parking spaces | 40 | 18 | 62.3 |
| YOLOv5 + SPPF | zebra crossings | 17 | 1 | 89.1 |
| | bus stations | 5 | 0 | 80.3 |
| | roadside parking spaces | 29 | 29 | 68.7 |
| YOLOv5 + K-means++ | zebra crossings | 18 | 0 | 88.4 |
| | bus stations | 4 | 1 | 81.7 |
| | roadside parking spaces | 37 | 21 | 73.3 |
| YOLOv5 + ASFF | zebra crossings | 18 | 0 | 88.6 |
| | bus stations | 4 | 1 | 84.0 |
| | roadside parking spaces | 44 | 14 | 71.8 |
| ASFF-YOLOv5 (YOLOv5 + ASFF + RFB) | zebra crossings | 18 | 0 | **89.5** |
| | bus stations | 5 | 0 | **89.7** |
| | roadside parking spaces | 50 | 8 | **80.3** |

Note: The bolded font is the best result for each. Number 1 is the number of road traffic elements that were correctly detected. Number 2 is the number of road traffic elements that were missed.

## 5. Discussion

Compared with other methods, the mAP of the proposed method had a great improvement, with an increase ranging from 0.3% to 39.2%, which verified that the proposed ASFF-YOLOV5 algorithm can effectively improve the detection accuracy of multielement road traffic. Different from other scholars' research [40–45], this paper considered multiscale object detection, especially in small object detection whose detection accuracy was substantially improved. Compared with the results of different network models, the detection accuracy of the proposed method was improved both in terms of the overall accuracy and individual objects for detection, which proves the superiority of the proposed method for multiscale object detection.

To verify the practicability and correctness of the proposed method, ablation experiments were conducted for different scenarios. The prediction results showed that the proposed method achieved the optimal detection accuracy compared with other methods. At the same time, the proposed method had the lowest number of missed detections, the highest detection accuracy, and the best detection effect compared with other methods for large complex scenes. This proves the practicality and superiority of the proposed method. It can be directly applied to image maps of large scenes and provides a more intelligent and convenient method for updating geographic information databases. However, in complex large scenes, there is inevitably the case of missed roadside parking space detections with occlusion since the semantic information of these parking spaces is more inconspicuous and the features are difficult to extract, which is also one of the challenges to be further addressed in subsequent work.

## 6. Conclusions

To address the problems of little data extraction of traffic element information, low automation but high demand, small element scale, and detection susceptibility to environmental interference, an ASFF-YOLOv5 algorithm for UAV remote sensing images was proposed in this paper. In this algorithm, an adaptive spatial feature fusion method based on a receptive field module is adopted to make full use of different scale information, improve the invariance of the feature scale, and improve the detection effect of small objects.

When detecting multiple road traffic elements, the mAP of the proposed method reached 93.1%, which is 19.2% higher than that of the original YOLOv5 network. A comparison experiment and ablation experiment proved that the proposed method could solve the problem of detection errors and missed detections for multielement road traffic, improve the detection accuracy of small and medium objects and intensive objects of multifactor road elements, and provide a new solution for the construction of basic geographic traffic information databases.

However, there are shortcomings of the proposed method. The experiments only focused on three types of elements, zebra crossings, bus stations, and roadside parking spaces. Subsequent work will expand the dataset to complete the automatic recognition and detection of more elements. At the same time, there are also missed detections of obscured small objects, and subsequent work will further improve and enhance the detection accuracy of small objects.

**Author Contributions:** M.Q.: algorithm proposal and testing, data processing, manuscript writing, and research conceptualization. L.H.: funding acquisition, directing, and manuscript review. B.-H.T.: project administration and supervision. All authors have read and agreed to the published version of the manuscript.

**Funding:** This research was funded by National Natural Science Foundation of China (41961039), Yunnan Fundamental Research Projects (grant NO. 202201AT070164 and 202101AT070102).

**Data Availability Statement:** The original data have not been made publicly available, but it can be used for scientific research. Other researchers can send emails to the first author if needed.

**Conflicts of Interest:** The authors declare no conflict of interest.

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
