# Peer review of "ASFF-YOLOv5: Multielement Detection Method for Road Traffic in UAV Images Based on Multiscale Feature Fusion"

_remotesensing, doi:10.3390/rs14143498_

Round 1
Reviewer 1 Report
The authors responded to my suggestions and comments appropriately. Therefore, I have no further suggestions or comments.
Author Response
Once again, thank you very much for your comments and suggestions.
Reviewer 2 Report
This version is a substantial improvement over the original. Some thought may be given to condense it further but it reads clearly the way it is.
I suggest that you define "small" and "medium" objects in terms of approximate actual dimensions. This would give a reader a sense of scale. Also, it would be very useful to stipulate the minimum object size that the proposed algorithm would be capable of detecting and classifying. For example, can we use to detect local pavement distress?
I think that the last sentence of the abstract (A comparison experiment.....) fits better as a conclusion. I would place it in the conclusions section instead of the sentence starting with "Experimental results verify..." (line 623).
Author Response
Please see the attachment.

This manuscript is a resubmission of an earlier submission. The following is a list of the peer review reports and author responses from that submission.
Round 1
Reviewer 1 Report
I think there is enough novelty in the methodology. But there is need for rewriting, especially the introduction, to introduce all methodologies and previous studies on Yolo-based small object detection, and then justifying this study. Also thoroughly referencing the previous study from the authors and considering the reference methods needed in this study.
Huang, L.; Qiu, M.L.; Xu, A.Z; Sun, Y.; Zhu, J.J. UAV imagery for automatic multi-element recognition and detection of road traffic elements. Aerospace 2022, 9, 198.
Reviewer 2 Report
It is not clear what the ultimate objective of this research is. The authors are obviously experts in the area of feature detection but the application to road traffic may not be optimal. The manuscript is unnecessarily long. It should be condensed in such a way that a reader understands fully the contribution to the state of the practice. Some of the items that need attention are as follows:
- Many figures are just too small to read. They need to be expanded or redrawn. Examples include Figs. 3, 4, 5, 6.
- What is the sensor used for image acquisition? What are its specifications?
- What are we trying to detect? Is is a particular traffic object or a potential defect in some object?
- What is the minimum object size that can be classified?
The authors conclude that the proposed method can be used to detect zebra crossings, bus stations, and roadside parking places. These features are usually fixed location with geographic coordinates known a priori. If we know where they are then what are we really detecting?
In summary I think the methodology proposed by the authors may have some specific practical applications but it is not evident that road traffic or ITS is one of them. Also, why is it focused exclusively on UAV?
Reviewer 3 Report
In line #41 - 42, the author stated that there are few studies on extraction of road traffic multi-element information. I suggest that the author cite some of these studies as they are more related to the author's study.
In line #404 and 405, the author listed some of the netoworks considered for comparative study. On what bases these networks are selected out of the many networks available? It would be good if the author states why these are selected.
In tables such as 3, 5, 6 & 7, it would better if the author replaces the phrase "Proposed method" by the acronomy such as ASFF-YOLOv5.